# Measuring similarity between embedding spaces using induced neighborhood graphs

## Abstract

Deep Learning techniques have excelled at generating embedding spaces that capture semantic similarities between items. Often these representations are paired, enabling experiments with analogies (pairs within the same domain) and cross-modality (pairs across domains). These experiments are based on specific assumptions about the geometry of embedding spaces, which allow finding paired items by extrapolating the positional relationships between embedding pairs in the training dataset, allowing for tasks such as finding new analogies, and multimodal zero-shot classification. In this work, we propose a metric to evaluate the similarity between paired item representations. Our proposal is built from the structural similarity between the nearest-neighbors induced graphs of each representation, and can be configured to compare spaces based on different distance metrics and on different neighborhood sizes. We demonstrate that our proposal can be used to identify similar structures at different scales, which is hard to achieve with kernel methods such as Centered Kernel Alignment (CKA). We further illustrate our method with two case studies: an analogy task using GloVe embeddings, and zero-shot classification in using CLIP and BLIP-2 embeddings. Our results show that accuracy in both analogy and zero-shot classification tasks correlates with the embedding similarity. These findings can help explain performance differences in these tasks, and may lead to improved design of paired-embedding models in the future.

## 1 Introduction

Several tasks in machine learning rely on obtaining pairs of corresponding representations from the same item in the same or different domains, such as word pairs in an analogy task (where the underlying item is the analogy itself) or items from different media or text modalities, such as text-to-image cross-modal representations. Each item representation is usually converted to an embedding vector in a space associated with its respective domain. Higher performances in these paired-item tasks are often linked to the preservation of proximity between embeddings related to the same item as we cross from one space of embeddings to the other. Thus, the study of how proximity is preserved between spaces may provide deeper insights into the reasons underlying performance differences in these tasks.

**Contributions**[1] In this work, we introduce a method, namely Nearest Neighbor Graph Similarity (NNGS), to measure the similarity between paired embeddings of items based on the estimated Jaccard similarity between the neighborhoods of induced graphs (Donnat and Holmes, 2018). We show that the neighborhood size can be adjusted so that it reflects the locality of changes observed in the similarity measure. In two case studies, we show that the proposed metric can provide insight on the reasons underlying performance in both the analogy and the zero-shot classification tasks. In special, it can be used to identify categories of words that are not represented adequately for the analogy task, and to identify prompt templates that can lead to higher accuracy in zero-shot classification.

---

[1]Code for the method and for the experiments in this paper can be found at:
`https://anonymous.4open.science/r/graph_structure-F301` and
`https://anonymous.4open.science/r/graph_structure-F301/experiments/nngs_paper/README.md`.

## 2 RELATED WORK

### 2.1 EMBEDDINGS

**Semantic Word Embeddings** Prior research involving embedding spaces has found usefulness in particular topological characteristics, such as the one spanned by GloVe (Pennington et al., 2014). In GloVe, each word from a vocabulary is represented by a vector $e_{\text{word}}$ in a so-called *semantic* space. Within a semantic space, a change of meaning is represented by a translation vector, that is, stating that the relationship between "a" and "b" is the same as that between "c" and "d" means that their embeddings have the relationship:

$$e_b - e_a = e_d - e_c = e_{\text{category A} \to \text{category B}}. \tag{1}$$

In Equation 1, $e_{\text{category A} \to \text{category B}}$ is a vector that maps elements from one category to another (e.g., countries to capitals, adjectives to corresponding superlatives, etc.) (Ethayarajh et al., 2018).

**Cross-modal Embeddings** More recently, multimodal embedding spaces have been used to represent items according to their content in different modalities. In CLIP (Radford et al., 2021), for example, the image corresponding to a photo of a car is represented by a vector $e_i$, the text "a photo of a car" is represented by $e_t$ and, ideally, $e_i \approx e_t$. In BLIP-2 (Li et al., 2023), the multimodal models are obtained by training unimodal language models that map onto each other using an attention mechanism. These ideas allow zero-shot classification by calculating the embeddings of an unknown image and comparing it to the embeddings of several candidate texts, and then choosing the most similar text as the class for that image.

**Assessing embeddings** Traditionally, the properties of embedding spaces have been evaluated using accuracy in task-specific benchmarks (Faruqui et al., 2016). This is important, as it allows a straightforward comparison between systems and a direct measure of its utility, but leaves gaps that were only addressed later, like the requirement for exponential data (Udandarao et al., 2024) or the tendency to behave like a simple bag-of-words (Yuksekgonul et al., 2023) in cross-modal zero-shot learning and the propensity to biasing embedding spaces according to term frequencies (Faruqui et al., 2016). Importantly, in subsequent work, these embeddings have been analyzed to find properties that could help explain their performances, such as studies on bounds for linear analogies (Ethayarajh et al., 2018) in GloVe and the properties of partial orthogonality (Jiang et al., 2023) and linear compositionality in CLIP (Trager et al., 2023).

### 2.2 SIMILARITY BETWEEN REPRESENTATIONS

Items from a dataset can have different representations. The similarity between representations can be measured with a *similarity index* $s(X, Y)$, which compares different representations $X \in \mathbb{R}^{n \times p_1}$ and $Y \in \mathbb{R}^{n \times p_2}$ (with possibly different dimensionalities $p_1$ and $p_2$, with $p_1 \leq p_2$ without loss of generality) of the same set of $n$ items (Kornblith et al., 2019).

Arguably, a similarity index should be invariant to orthonormal linear transformations and isotropic non-zero scaling and translation (Kornblith et al., 2019). More generally, a similarity index could be invariant to general invertible linear transformations (Raghu et al., 2017), but this property will not be adopted in the present work as it implies that all datasets have the same similarity index if $p_1 > n$ (Kornblith et al., 2019).

An important measure for similarity between representation is the Centered Kernel Alignment (CKA) measure (Kornblith et al., 2019), calculated by:

$$\text{CKA}(X, Y) = \frac{\text{tr}(K_x^c(X^c)K_y^c(Y^c))}{\sqrt{\text{tr}(K_x^c(X^c)K_x^c(X^c))\text{tr}(K_y^c(Y^c)K_y^c(Y^c))}}, \tag{2}$$

where $\text{tr}(.)$ is the trace operator, $K_x$ and $K_y$ are kernel functions applied to $X$ and $Y$, and the $^c$ operation consists of centering a cloud point by subtracting the dimension-wise mean from all elements.

CKA draws from the idea of assessing how distances between items change from one representation to another. With a linear kernel, a changes in the distance between data blobs have a larger impact

than the distance between points within each blob, because distance changes are usually larger. By using an RBF kernel $(\exp -D(X)/2\sigma^2$, where $D(X)$ is a matrix with pairwise distances between the items of $X$) it is possible to give more weight to changes made on smaller distances.

However, the weighting process is distance-based, that is, the choice of an adequate value for $\sigma$ depends on the radius of the localities being analyzed. This value is usually hard to consistently obtain, as it can change even if data is subject to isotropic transformations. Also, it can have little meaning if data is spread in irregular manifolds.

These aspects do not necessarily constitute limitations of CKA. Rather, they are a consequence of the definition of similarity used to build the metric. A different approach, called GULP (Boix-Adsera et al., 2022), defines similar representations as those that lead to the same predictive performance for a set of items, which has shown increased results, when compared to CKA, in the task of predicting neural network layers using the CIFAR-10 dataset (Krizhevsky and Hinton, 2009). It is define in terms of the centralized representations $X^c$ and $Y^c$ as:

$$
\begin{aligned}
\hat{\Sigma}_X &= \tfrac{1}{n}\sum_{i=1}^N (X^c)^T X^c \\
\hat{\Sigma}_Y &= \tfrac{1}{n}\sum_{i=1}^N (Y^c)^T Y^c \\
\hat{\Sigma}_{XY} &= \tfrac{1}{n}\sum_{i=1}^N (X^c)^T Y^c \\
\hat{\Sigma}_X^{-\lambda} &= (\hat{\Sigma}_X + \lambda I)^{-1} \\
\hat{\Sigma}_Y^{-\lambda} &= (\hat{\Sigma}_Y + \lambda I)^{-1} \\
\mathrm{GULP}(X,Y,\lambda) &= \mathrm{tr}(\hat{\Sigma}_X^{-\lambda}\hat{\Sigma}_X\hat{\Sigma}_X^{-\lambda}\hat{\Sigma}_X) + \mathrm{tr}(\hat{\Sigma}_Y^{-\lambda}\hat{\Sigma}_Y\hat{\Sigma}_Y^{-\lambda}\hat{\Sigma}_Y) - 2\mathrm{tr}(\hat{\Sigma}_X^{-\lambda}\hat{\Sigma}_{XY}\hat{\Sigma}_Y^{-\lambda}\hat{\Sigma}_{XY}^T),
\end{aligned}
\tag{3}
$$

where $\lambda$ is a regularization parameter.

Another definition of similarity, namely ContraSIM (Rahamim and Belinkov, 2023), relies on the idea that it should be possible to map similar representations to a common embedding space using contrastive loss. Thus, ContraSIM (Rahamim and Belinkov, 2023) uses the loss of such mapping as a similarity measure. ContraSIM was evaluated using images from the CIFAR-10 (Krizhevsky and Hinton, 2009) and CIFAR-100 (Krizhevsky et al.) datasets.

We observe that both GULP and ContraSIM are extensions of the Canonical Correlation Analysis (CCA) (Hotelling, 1992), as they measure the error obtained due to projecting the representations $X$ and $Y$ onto a common space. ContraSIM (Rahamim and Belinkov, 2023) allows greater flexibility in this projection by allowing the use of arbitrary encoders, but this also implies in the need to train an encoder. Training contrastive losses implies in a series of particular problems and, because of that, ContraSIM was not used in further experiments.

### 2.3 STRUCTURAL SIMILARITY BETWEEN GRAPH NODES

Given a point cloud $V = \{v_i\}$, $v_i \in \mathbb{R}^d$, a directed graph $G_k = (V, E_k)$ can be induced by connecting each point to its $k$ closest neighbors (Eppstein et al., 1997), $k \in [1, n-1]$, thus forming the set of directed edges $E_k$. A point is not allowed to have itself as a neighbor, thus $G_k$ is a directed simple graph. We say that $G_k$ is induced by $k$-neighborhood in $V$. The set of neighbor indices of a vertex $v_i$ is $N_k(v_i) = \{j | (i, j) \in E_k\}$. Two graph vertices $v_i$ and $v_j$, $i \neq j$, can be said to be structurally equivalent if they share the same set of neighbors, that is, $N(v_i) = N(v_j)$ (Newman, 2018). If they only share some neighbors, we can use the Jaccard similarity $J(N(v_i), N(v_j))$ (Equation 4) to measure the degree of structural similarity between $v_i$ and $v_j$ (Donnat and Holmes, 2018):

$$
J(N_k(v_i), N_k(v_j)) = \frac{|N_k(v_i) \cap N_k(v_j)|}{|N_k(v_i) \cup N_k(v_j)|}.
\tag{4}
$$

A soft approach to graph structural similarity has shown to lead to minor accuracy increases in downstream classifiers based on image-text pre-trained embeddings (Sobal et al., 2024). In this work, we further assess how this idea can relate to zero-shot learning tasks in this same scenario. Also, we bring evidence that this type of approach can relate to task performance in other tasks such as an embedding-space analogy task.

# 3 PROPOSED METHOD: NEAREST NEIGHBORHOOD GRAPH SIMILARITY

In this section, we propose Nearest Neighborhood Graph Similarity (NNGS). We discuss some some theoretical perspectives that explain how to interpret its results.

## 3.1 SIMILARITY BETWEEN CORRESPONDING POINT CLOUDS

The notion of structural similarity between nodes (Equation 4) can be adapted to compare point clouds where there is a one-to-one correspondence between points in both clouds[2]. For such, let $X = \{x_i\}$ and $Y = \{y_i\}$, $i \in \{1, 2, \cdots, n\}$, be point clouds where $x_i \in \mathbb{R}^{p_1}$ and $y_i \in \mathbb{R}^{p_2}$ are corresponding points. Two graphs, $G_{X,k} = (X, E_{X,k})$ and $G_{Y,k} = (Y, E_{Y,k})$, can be induced by $k$-neighborhood on $X$ and $Y$ as described previously.

Let $N_{X,k}(x_i)$ be the set of indexes of the $k$ points closest to $x_i$ in $X$, and define $N_{Y,k}(y_i)$ analogously. Importantly, the distance measure used for this operation can be selected according to the problem. The structural similarity between corresponding points $x_i$ and $y_i$ is defined as:

$$J(N_{X,k}(x_i), N_{Y,k}(y_i)) = \frac{|N_{X,k}(x_i) \cap N_{Y,k}(y_i)|}{|N_{X,k}(x_i) \cup N_{Y,k}(y_i)|}. \tag{5}$$

Note that, different from Equation 4, points $x_i$ and $y_i$ do not belong to the same graph, hence the need for comparing their sets of neighbor *indices*, and not the neighboring points themselves.

From Equation 5, we define the Nearest Neighborhood Graph Similarity (NNGS) between point clouds $X$ and $Y$ as the average structural similarity between corresponding points:

$$\text{NNGS}(X, Y, k) = \frac{1}{n} \sum_{i=1}^{n} J(N_{X,k}(x_i), N_{Y,k}(y_i)). \tag{6}$$

Note that the structural similarity between point clouds depends only on their neighborhood structure: the precise definitions of distance in each domain, or the value of point coordinates only impact the structural similarity if they change the ranks of closest points among the cloud. For instance, if the point clouds are equal, that is, $X = Y$, then $\text{NNGS}(X, Y, k) = 1, \forall k \in [1, n - 1]$. Furthermore, $\text{NNGS}(X, Y, k) = \text{NNGS}(X, Y', k)$ if $Y'$ is constructed from $Y$ by applying only isotropic scaling, translations, or orthonormal transformations, as they not change the ranks of neighborhood distances between points.

## 3.2 PROPERTIES OF NNGS

In addition to the invariance to isotropic scaling, translations, and orthonormal transformations, NNGS has important properties related to its behavior regarding white noise, and invariance properties related to the number $n$ of points in the evaluated set and their dimensionality ($p_1$ and $p_2$). These properties are discussed next. In the following experiments, we used the Cosine similarity to define neighborhoods.

### 3.2.1 SIMILARITY OF RANDOM POINT CLOUDS

If $X = \{x_i\} \overset{\text{iid}}{\sim} \mathcal{N}(0, I_{d_X}), Y = \{y_i\} \overset{\text{iid}}{\sim} \mathcal{N}(0, I_{d_Y})$, then the neighborhoods $N_{X,k}(x_i)$ and $N_{Y,k}(y_i)$ become random draws. In this case, the intersection cardinality $|N_{X,k}(x_i) \cap N_{Y,k}(y_i)|$ for a randomly chosen $i$ follows a hypergeometric distribution with $n - 1$ total elements, $k$ elements of interest and $k$ draws. Therefore, $J(N_{X,k}(x_i), N_{Y,k}(y_i))$ is a random variable whose expected value over all possible pairs of point clouds is (see proof in Appendix B):

$$\mathbb{E}[J(N_{X,k}(x_i), N_{Y,k}(y_i)] \geq H(k) = \frac{k}{2(n - 1) - k}. \tag{7}$$

Importantly, $\mathbb{E}[\text{NNGS}(X, Y, k)] \geq \mathbb{E}[J(N_{X,k}(x_i), N_{Y,k}(y_i)]$ in this case because the neighborhoods between points used to calculate $J$ in Equation 5 are not independent. Rather, the choice of the first

---

[2]This situation can be found in multimodal datasets used to train, e.g., CLIP.

neighborhoods impact the remaining ones by restricting the number of possible choices that maintain the symmetry property of distances (see Appendix C for a counter-example of independence). Hence, $H(k)$ can be seen as a lower bound for $\mathbb{E}[\text{NNGS}(X, Y, k)]$ if point clouds $X$ and $Y$ are i.i.d.

In most practical cases, however, the point clouds $X$ and $Y$ are neither equal nor entirely independent. These cases are further discussed next.

### 3.2.2 SIMILARITY IN THE PRESENCE OF NOISE

One way to progressively distort the structure of a point cloud is by adding white noise. For such, we generate a point cloud $X = \{x_i\} \overset{\text{iid}}{\sim} \mathcal{N}(0, I)$. After that, we define a desired Signal-to-Noise Ratio (SNR) and calculate $Y = \{y_i\} = \{x_i + \alpha \phi_i\}$, where $\{\phi_i\} \overset{\text{iid}}{\sim} \mathcal{N}(0, I)$ and $\alpha = 10^{-\frac{\text{SNR}}{20}}$.

With low noise, SNR $\to \infty$, $\alpha \to 0$ and $X \to Y$, thus $\text{NNGS}(X, Y, k) \to 1$. However, in a very high noise scenario, SNR $\to -\infty$, causing $X$ and $Y$ to become uncorrelated, thus $\text{NNGS}(X, Y, k) \to H(k)$. As shown in Figure 1, intermediate SNR values lead to curves that are between the other two, but are clearly not simple linear interpolations of the extreme values.

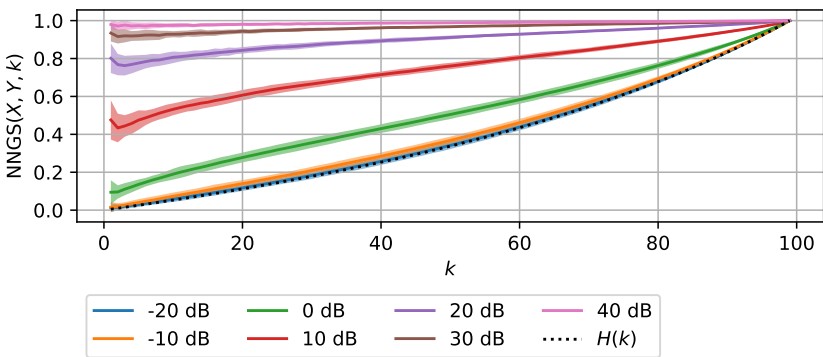

Figure 1: Mean structural similarity in the presence of white noise. The point clouds were generated using $n = 100$ points and $d = 50$ dimensions. The impact of changing $n$ and $d$ are respectively discussed in Section 3.2.3 and Section 3.2.4. The curves were bootstrapped as to obtain mean and standard deviation, and the depicted intervals correspond to two standard deviations. With very low SNRs, $\text{NNGS}(X, Y, k) \to H(k)$, while in high SNRs $\text{NNGS}(X, Y, k) \to 1$. An SNR sweep is provided in Appendix D.

Importantly, lower values of $k$ refer to a close vicinity, whereas larger values of $k$ make $\text{NNGS}(X, Y, k)$ refer to a more general vicinity in the point clouds. In Figure 1, the similarity curves do not cross, indicating that the similarity is equally distorted in all vicinity sizes. Although this suggests that any value of $k$ could be equally useful to evaluate $\text{NNGS}(X, Y, k)$, the non-crossing behavior is a property specific to distortions caused by additive random noise.

### 3.2.3 CHANGING THE POINT CLOUD SIZE $n$

In Equation 7, if $c = k/(n-1)$, then:

$$H(k) = \frac{k}{2(n-1) - k} = \frac{\frac{k}{n-1}}{2 - \frac{k}{(n-1)}} = \frac{c}{2 - c}, \tag{8}$$

thus, $H(k)$ only depends on the relative neighborhood size $c = k/(n-1)$.

Figure 2 shows that $\text{NNGS}(X, Y, k)$ does not change with $n$ if $c$ is kept constant, that is, $k = \lfloor c(n-1) \rfloor$ for each $n$. This observed behavior allows comparing $\text{NNGS}(X, Y, k)$ between experiments with different point cloud size $n$ as long as the corresponding values of $c = k/(n-1)$ are close in value.

In this case, using small values for $c$ makes $\text{NNGS}(X, Y, k)$ focus on a local neighborhood, whereas values of $c$ closer to 1 shift $\text{NNGS}(X, Y, k)$ towards assessing a more global structure.

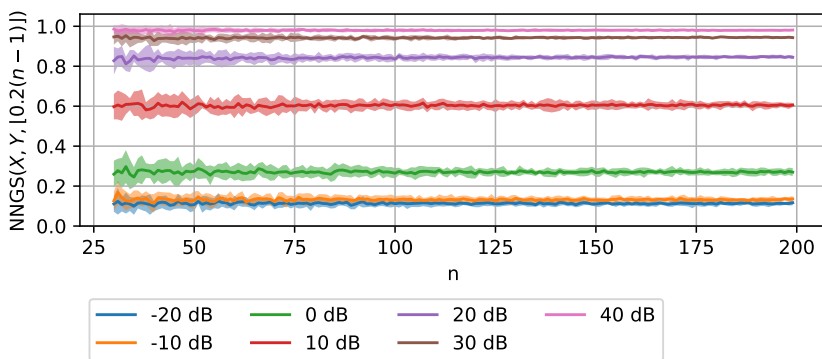

Figure 2: The similarity $\text{NNGS}(X, Y, k)$ remains nearly constant as the point cloud size $n$ increases as long as $k = \lfloor c(n-1) \rceil$ for a constant value of $c$. In this figure, we arbitrarily chose $c = 0.2$.

### 3.2.4 INVARIANCE TO POINT CLOUD DIMENSIONALITY ($p_1$ AND $p_2$)

We also investigate how $\text{NNGS}(X, Y, k)$ is affected when the point cloud dimensionality ($p_1$ and/or $p_2$) is changed. When the dimensionality increases, it is more likely that the cosine similarity between random points becomes equal to zero, that is, all points are orthogonal to each other. Thus, adding noise with a particular variance has an equal chance of taking a point to a different neighborhood. Consequently, as shown in Figure 3, the dimensionality of the point clouds do not impact $\text{NNGS}(X, Y, k)$.

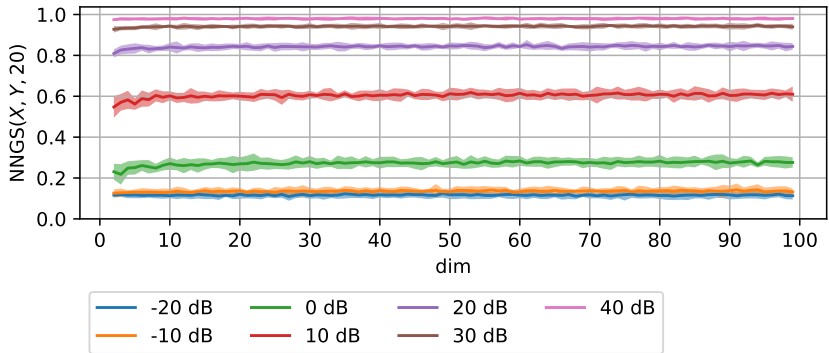

Figure 3: The similarity $\text{NNGS}(X, Y, k)$ is not affected by the increase of dimensionality, except in very low dimensionality settings (less than 10). In this figure, we fixed $k = 20$ and the point cloud size $n = 100$. This is a consequence of the fact that random points in higher dimensionalities are likely to be orthogonal to each other, thus the noise variance required to get closer to different neighbors is similar regardless of the increased dimensionality.

This further indicates a consistency of this measure when evaluating point clouds in different dimensionalities - for example, assessing the distortions caused by non-linear projections in multilayer neural networks.

### 3.3 COMPARING NNGS, CKA, AND GULP

NNGS has two additional parameters when compared to CKA: the neighborhood size $k$, and the distance metric used to induce the neighborhood graph. GULP has one additional parameter: the regularization term $\lambda$. In datasets with more than one cluster, these parameters can be manipulated to find similarities at different scales and in different situations. The experiments below were conducted with toy datasets crafted especially to demonstrate these differences, as follows.

**Data representations with blobs** The latent representation of data points is frequently consisted of blobs, especially in the context of supervised classifiers. In this context, each blob corresponds to a different class. As will be discussed next, NNGS, CKA, and GULP behave differently depending on the nature of transformations applied to blobs. The algorithm and parameters used to create the data blobs are shown in Appendix E. The results for each experiment are shown in Table 1 and discussed next.

| Method | CKA | CKA RBF | CKA RBF | NNGS | NNGS | GULP | GULP |
|---|---|---|---|---|---|---|---|
| Kernel / distance | Linear | RBF | RBF | Minkowski | Minkowski | - | - |
| Parameter | - | $\sigma = 0.01$ | $\sigma = 3$ | $k = 5$ | $k = 300$ | $\lambda = 0.01$ | $\lambda = 1$ |
| Blobs with Different Scales | 0.86 | 1.00 | 0.98 | 1.00 | 0.76 | 15.43 | 0.11 |
| Unbal. Blobs w/ Dif. Scales | 0.10 | 1.00 | 0.31 | 1.00 | 0.99 | 1.50 | 0.02 |
| Noise Within Blobs | 0.99 | 1.00 | 0.96 | 0.03 | 0.99 | 6.88 | 0.02 |
| Shuffled Blobs | 0.16 | 1.00 | 0.84 | 1.00 | 0.63 | 2.07 | 0.49 |

Table 1: Similarity between artificial aligned datasets measured by variants of CKA, NNGS, and GULP. By changing $k$ in NNGS, it is possible to change the locality in which the similarity is measured. However, this is harder to obtain by changing the $\sigma$ value in CKA with an RBF kernel or the $\lambda$ parameter in GULP even if they follow the exact parameters used in the dataset creation.

**Blobs with different scales** A particularly interesting case is that in which representation $X$ is consisted of two data blobs and they are scaled differently to obtain $Y$. We observe that RBF CKA($\sigma = 0.1$) and NNGS($k = 5$) find a similarity of 1, as blobs are locally unchanged. NNGS($k = 300$) finds a small modification in the inter-cluster neighborhoods, whereas Linear CKA is slightly affected by the asymetric scaling. GULP assumes values between 0.11 and 15.43 depending on the regluarization factor $\lambda$, which makes it hard to interpret what is a high or a low value in the context of this measure.

**Unbalanced blobs with different scales** The same asymetric scaling used above can be applied when blobs have an unbalanced number of items. In this case, The average distance modification within the dataset is not preserved, whereas the closest neighborhoods are preserved. Compared to the balanced experiment, NNGS for a local neighborhood remains unchanged, whereas both Linear and RBF CKA for a more global neighborhood are affected. We observe that NNGS for a broader neighborhood is changed due to the modifications in the blob size. GULP presents low values, indicating a behavior similar to Linear CKA.

**Noise within blobs** Conversely, if an average amount of noise is added to $X$ so that $Y$ preserves the blob centroids, but shuffles the local neighborhood, CKA returns a high similarity between both representations, thus failing to identify the local noise due to the preservation of the higher-level similarity. GULP with a high $\lambda$ seems to be inaffected, whereas a low $\lambda$ lead to an increased value of similarity. Conversely, NNGS with a low $k$ identifies that local neighborhoods were modified. Yet, NNGS with a high $k$ is able to identify that blob neighborhoods were changed.

**Shuffling blob centroids** If data is consisted of blobs, it is possible to transform the representation $X$ by applying a translation to each blob centroid, but without changing relative positions within centroids, generating $Y$. In this case, long-range distances are highly impacted, whereas local neighborhoods are preserved. Consequently, CKA and GULP find a low similarity value, while NNGS with a low value for $k$ identifies that blobs are locally unchanged.

These experiments show that changing the value of $k$ implies in a modification of the locality of the similarity measure. Although theoretically the value of $\sigma$ in CKA with an RBF kernel plays the same role, it is easier to find suitable values for $k$ than to $\sigma$. In special, we observe that even very small values of $\sigma$ were innefective to find local changes in the point clouds.

In addition, we find that the values yielded by GULP are hard to classify as "high" or "low" as they can be unbounded and highly dependent on the regularization parameter $\lambda$.

The adequate value of $k$ can be obtained by standard techniques. These can be either observing datasets properties, like the number of expected points per data blob, or observing data plots that highlight data clusters. The value of $k$ implies in the type of similarity (local or global) being measured.

# 4 CASE STUDIES

The experiments described in this section[3] aim to demonstrate how the similarity measure $NNGS(X, Y, k)$ (Equation 6) can be used to unveil properties of embedding spaces. The first experiment use GloVe embeddings (Section 4.1) to find an association between the accuracy in analogy tasks and their corresponding similarity $NNGS(X, Y, k)$. The second experiment (Section 4.2) uses CLIP to generate text embeddings for candidate captions in a zero-shot classification task, and highlights that the accuracy in this task are associated with the structural similarity $NNGS(X, Y, k)$ between the caption embeddings and the image embeddings.

## 4.1 SIMILARITY IN SEMANTIC WORD EMBEDDINGS

Equation 1 suggests that analogies between groups of items can be represented by a translation. Thus, point clouds that are adequate for an analogy task based on Equation 1 are likely to present a high structural similarity as calculated by Equation 6.

We evaluate this assumption using the 300-dimensions GloVE embeddings and the analogy tasks defined in (Mikolov et al., 2013)[4]. For each analogy task, we find the two related point clouds (for example: one containing embeddings for countries and the other containing embeddings for capitals), and calculate the similarity between them using Equation 6 for various values of the relative neighborhood size $c$. The results are shown in Figure 4.

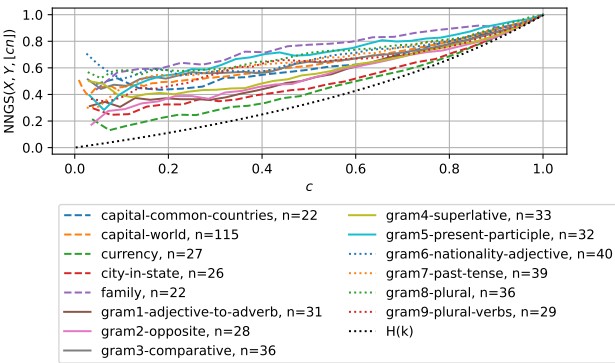

Figure 4: Mean structural similarity in GloVe embeddings for each analogy task. The curves present a shape similar to those in Figure 1, and some tasks clearly have greater similarity than others.

The curves present a behavior similar to those in Figure 1, but are more jittery, causing close curves to cross at various points. This is because the associated point clouds are related by transforms that allow Equation 1 to work, but also by non-uniform distortions due to the process of building the embeddings. Next, we evaluate the association between the structural similarity and the accuracy of each analogy task.

As shown in Figure 5, tasks in which $NNGS(X, Y, k)$ are higher tend to have higher accuracy. We observe a trend, as supported by the Pearson correlation between NNGS and the analogy accuracy (Pearson's $\rho = 0.86, p < 10^{-4}$). Also, we observe that this trend seems to diminish for high analogy accuracy values, which could be caused by the saturation of the accuracy measure.

Importantly, the same experiment using CKA presents, as well, a high correlation, whereas GULP is unable to capture this trend. This is should not be confused with a flaw of GULP; rather, it is an indication that the aspects that lead to a higher accuracy in the analogy task are unrelated to the

---

[3]The experiments were executed in an on-premises server. The experiments for the analogies task used a single-core CPU with 256 GB RAM: it did not require a GPU and most of the execution time was spent loading embeddings into memory. The CLIP model, used in the zero-shot-classification task, executed in a server with 256GB RAM, using a single NVIDIA Quadro RTX 6000 GPU with 24GB RAM. Experiments took around 1h in total.

[4]The repository that contains this data (`https://github.com/nicholas-leonard/word2vec/`) is licensed under the Apache licence

underlying ideas of GULP (similarity in the prediction accuracy). Hence, all three measures are useful to draw a complete diagnosis of this behavior.

## 4.2 SIMILARITY IN CROSS-MODAL ZERO-SHOT LEARNING

In a cross-modal zero-shot classification task, the text embeddings are calculated from a set of captions that are constructed from the class labels and template phrases. In CLIP, these templates are variations of the phrase "a [description] [photo/image] of X"[5], where "X" is substituted by the desired label.

We perform experiments to draw insight about the relationship between the phrasing of text captions, the zero-shot classification performance, and the similarity of cross-modal embeddings using the pre-trained CLIP model available on HuggingFace[6]. In these experiments, we assess the similarity between embeddings in each modality in the CIFAR-100 dataset (Krizhevsky, 2009; Krizhevsky et al.)[7] and the ImageNet dataset (Deng et al., 2009) using $\text{NNGS}(e_i, e_t, k)$. For such, we calculate the mean image embedding for each class ($e_i$). Then, we use the corresponding text embeddings $e_t$ to estimate $\text{NNGS}(e_i, e_t, k)$.

We also experiment with two methods to make new templates. The first is to change the original templates to their negations, that is, we use variations of "not a [description] [photo/image] of X". The second method is to make templates from phrases unrelated to images or photos using variations of famous film and music phrases. The complete list of prompts can be found in Appendix F.

In this situation, there are 100 points, but there are no blobs. In this experiment, we are interested in low neighborhood sizes, as they already relate to inter-class positions. For this reason, we use $k = 3$.

Next, we evaluate the zero-shot classification accuracies using each single template, and compare them with its $\text{NNGS}(X, Y, k)$ for $k = 3$, as shown in Figures 6 and 7. We present the measures $\rho_1$, which is the Pearson's correlation between NNGS and the zero-shot accuracy for each prompt template considering only the original templates, and $\rho_2$, which considers the added templates as well. We also present results of similar experiments using CKA and GULP.

Interestingly, results indicate that using negative templates instead of positive templates leads to accuracy differences of less than $0.02$. However, the film/music templates had a lower accuracy.

We find a high correlation between NNGS and the zero-shot classification accuracies, as shown in Figures 6 and 7. Importantly, the values of $\rho_2$ can be seen as skewed because the added prompts seem to form a series of poorly-performing elements. When only the original prompts are considered, the obtained values are lower, but still significant. However, we highlight that the same experiments, when performed with CKA or GULP, leads to a lower correlation, which indicates that NNGS was able to identify similarities that are more related to the zero-shot classification performance.

The plots in Figures 6a and 7a form a trend, suggesting that the similarity $\text{NNGS}(X, Y, k)$ is related to the zero-shot classification accuracy. However, we note that this trend cannot be observed if particular sets of templates are chosen, especially in some subsets of templates with high accuracy. This behavior could be due to reaching an accuracy higher limit, similarly to the observations in Figure 5.

We further experiment using the ImageNet embeddings generated by a BLIP-2 model (Li et al., 2023). In these experiments, we used the output of the [CLS] token as the text embedding, and the mean of all image token embeddings as the representative of each particular item. The results, shown in Figure 8, indicate that NNGS has a higher correlation with the task-specific performance.

Importantly, although NNGS yields a higher $\rho_1$ in BLIP-2 embeddings than in CLIP embeddings, the zero-shot classificatio accuracy is not necessarily higher. This means that NNGS correlates with the task-specific accuracy within the same embeddings, but not necessarily across different embedding

---

[5]The template list is available at `https://github.com/openai/CLIP/blob/main/data/prompts.md`.

[6]`https://huggingface.co/docs/transformers/v4.39.3/en/model_doc/clip`

[7]The CIFAR-100 website does not cite a specific licence for usage. However, this data is distributed under various packages and has been broadly used in research for benchmark purposes. We specifically use the distribution available through Pytorch.

domains. Also, we note that CKA and GULP have $\rho_2 < \rho_1$, which seems to be caused by the erratic behavior of movies/music-inspired prompts (green dots).

In all cases discussed in this section, the similarity $\text{NNGS}(X, Y, k)$ has shown to correlate with the task-specific accuracy value. In addition, $\text{NNGS}(X, Y, k)$ can be used to evaluate the underlying assumptions for each case ($e_b - e_a = e_d - e_c$ for the analogy task, and $e_t \approx e_i$ for the zero-shot classification task). Our results indicate that the more the underlying assumptions are met (as measured by $\text{NNGS}(X, Y, k)$), the greater is the system's tendency to reach higher performances.

## 5 CONCLUSION

Our study introduces a novel approach, namely Nearest Neighborhood Graph Similarity (NNGS) to evaluating the similarity of paired embedding spaces using GloVe, CLIP, and BLIP-2 models as case studies. The similarity is measured using the average Jaccard similarity between the corresponding nodes of k-neighborhood-induced graphs. The resulting measure informs how well these models preserve the underlying structural properties of the data. Our findings demonstrate a strong correlation between similarity and task-specific accuracy, indicating that maintaining structure within corresponding embedding spaces is a possible route for achieving high performance in tasks such as analogy calculation and cross-modal zero-shot classification.

We have compared our method with two previous approaches (CKA and GULP), showing that NNGS is invariant to dimensionality changes, robust to changes in the hyperparameter, and can be made invariant to point cloud sizes by using a relative neighborhood size instead of an absolute one. Also, we have shown that changing the value of $k$ can make NNGS focus on local or on global transformations over data, and finding a value for $k$ can be done by investigating the clusters found within the dataset. We have shown that a well-tuned NNGS presents a greater correlation with task-specific performances than CKA and GULP.

Importantly, the results shown here do not mean that NNGS is a superior similarity measure in all possible cases. Rather, we note that each representation similarity measure draws from different assumptions, hence is more adequate for a different situation. Consequently, different similarity metrics can be used harmoniously to provide different viewpoints about embedding spaces.

Looking forward, our findings raise important questions about other practical applications of similarity metrics. We surmise that they could serve as an effective stopping criteria in early-stopping strategies, aiding in the prevention of overfitting in cross-modal learning. Also, they could offer insights into identifying groups of words where analogy tasks excel, guiding future model development. Last, similarity could inform broader discussions on prompt engineering, helping to provide explanations regarding the underlying agents behind the performance of specific prompt templates.

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

# A    FIGURES WITH RESULTS

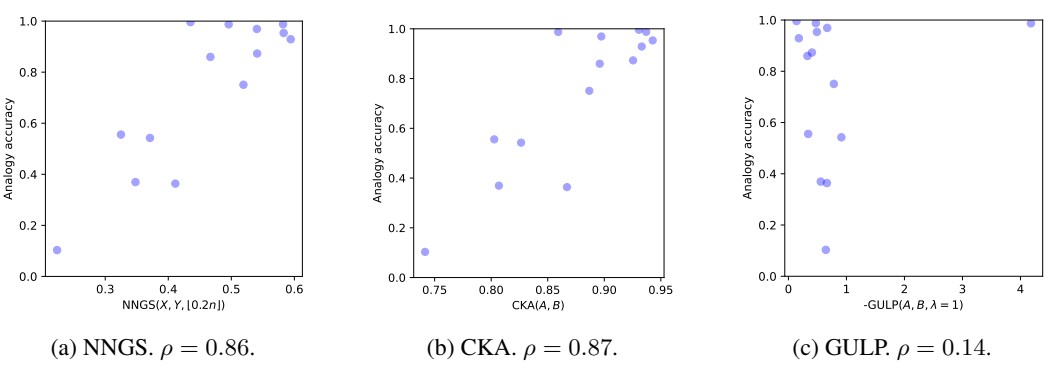

(a) NNGS. $\rho = 0.86$.          (b) CKA. $\rho = 0.87$.          (c) GULP. $\rho = 0.14$.

Figure 5: Pearson's correlation between NNGS, CKA, and GULP and the accuracy in the analogy task in GloVe embeddings. Each point represents an analogy task. The similarity measured by NNGS and CKA have a high correlation to the analogy accuracy, whereas this trend is unobserved in GULP.

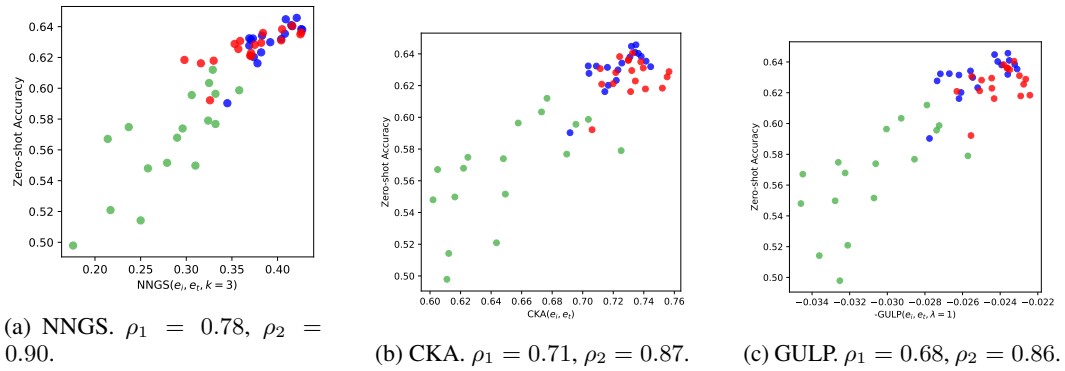

(a) NNGS. $\rho_1 = 0.78$, $\rho_2 = 0.90$.          (b) CKA. $\rho_1 = 0.71$, $\rho_2 = 0.87$.          (c) GULP. $\rho_1 = 0.68$, $\rho_2 = 0.86$.

Figure 6: Pearson's correlation between NNGS, CKA, and GULP and zero-shot accuracy in the CIFAR-100 dataset using CLIP embeddings. Each point represents one template (blue: original templates, red: negative templates, green: templates inspired in movies and music). We report $\rho_1$ related to using the original templates, and $\rho_2$, related to using the original and the added templates.

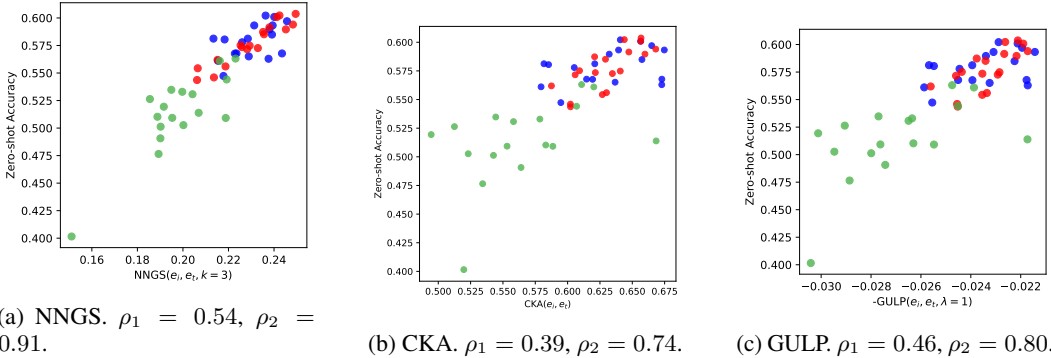

(a) NNGS. $\rho_1 = 0.54$, $\rho_2 = 0.91$.          (b) CKA. $\rho_1 = 0.39$, $\rho_2 = 0.74$.          (c) GULP. $\rho_1 = 0.46$, $\rho_2 = 0.80$.

Figure 7: Pearson's correlation between NNGS, CKA, and GULP and zero-shot accuracy in the ImageNet dataset using CLIP embeddings. Each point represents one template (blue: original templates, red: negative templates, green: templates inspired in movies and music). We report $\rho_1$ related to using the original templates, and $\rho_2$, related to using the original and the added templates.

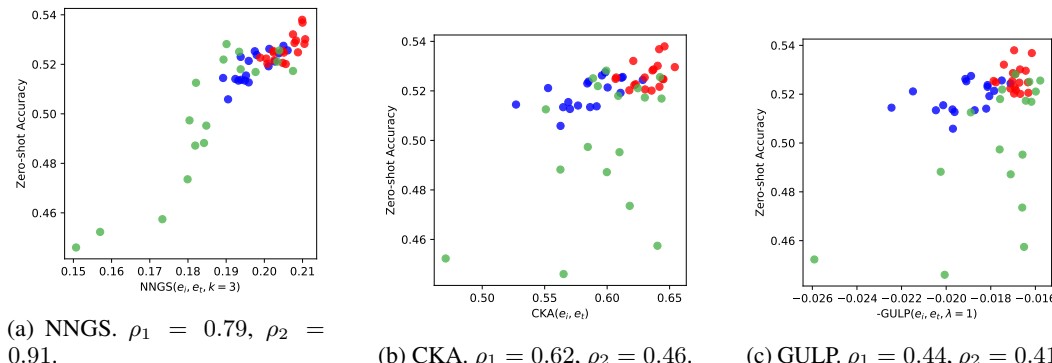

(a) NNGS. $\rho_1 = 0.79$, $\rho_2 = 0.91$.

(b) CKA. $\rho_1 = 0.62$, $\rho_2 = 0.46$.

(c) GULP. $\rho_1 = 0.44$, $\rho_2 = 0.41$.

Figure 8: Pearson's correlation between NNGS, CKA, and GULP and zero-shot accuracy in the ImageNet dataset using BLIP-2 embeddings. Each point represents one template (blue: original templates, red: negative templages, green: templates inspired in movies and music). We report $\rho_1$ related to using the original templates, and $\rho_2$, related to using the original and the added templates.

## B  PROOF OF EQUATION 7

Suppose we have a bag containing $M$ numbered balls, which are numbered from 1 to $M$. We build set $A$ by randomly drawing $k$ balls from the bag without replacement. Then, we put all balls back in the ball and build set $B$ by drawing the same number $k$ of balls without replacement. The question is: what is the cardinality of the intersection between $A$ and $B$?

### B.1  SOLUTION: HYPERGEOMETRIC DISTRIBUTION

After building set $A$, we can mark all balls with a cross before replacing them. Then, when building set $B$, we have the following situation:

- We have $M$ balls,

- There are $k$ balls of interest,

- There are $k$ draws without replacement

- $x = |A \cap B|$ corresponds to the number of successes within these draws.

This directly leads to a hypergeometric distribution:

$$p(x; M, k) = \text{Hyp}(x, M, k, k) = \frac{\binom{k}{x}\binom{M-k}{k-x}}{\binom{M}{k}} \tag{9}$$

And, following the properties of a hypergeometric distribution:

$$\mathbb{E}[x] = \frac{\# \text{ objects of interest} \times \# \text{ draws}}{\# \text{ total objects}} = \frac{k^2}{M} \tag{10}$$

### B.2  THE CASE OF JACCARD SIMILARITY

The Jaccard Similarity between sets $A$ and $B$, with $k$ draws, is:

$$J(A, B, k) = \frac{|A \cap B|}{|A \cup B|} = \frac{|A \cap B|}{|A| + |B| - |A \cap B|} = \frac{|A \cap B|}{2k - |A \cap B|}. \tag{11}$$

Let $x = |A \cap B|$ and consider the function:

$$f(x) = J(A, B, k) = \frac{x}{2k - x} \tag{12}$$

This function is convex, hence, by Jensen's inequality,

$$\mathbb{E}[f(x)] \geq \frac{\mathbb{E}[x]}{2k - \mathbb{E}[x]} \tag{13}$$

Substituting $x$ using Equation 10, we have:

$$\mathbb{E}[J(A, B, k)] \geq \frac{k^2/M}{2k - (k^2/M)} \tag{14}$$

We can simplify:

$$\frac{k^2/M}{2k - (k^2/M)} = \frac{k^2}{M(2k - (k^2/M))} = \frac{k^2}{2Mk - k^2}. \tag{15}$$

Since we know that $k$ is strictly positive ($k > 0$), then:

$$\mathbb{E}[J(A, B, k)] \geq \frac{k}{2M - k} \tag{16}$$

### B.3 CHOOSING $k$ AS A FRACTION OF $M$

If $k = cM, c \in \{i/M | 1 \leq i \leq M, i \in \mathbb{N}\}$, then we have:

$$\mathbb{E}[J(A, B, k)] \geq \frac{cM}{2M - cM} = \frac{cM}{(2 - c)M} = \frac{c}{2 - c} \tag{17}$$

## C  PROOF THAT NEIGHBORHOOD CHOICE IS NOT INDEPENDENT

Let $x_1, x_2, x_3 \in \mathbb{R}^d$ be 3 points in a point cloud. Using a distance measure $\gamma(x_i, x_j)$, we induce a k-neighbor graph with $k = 1$. Suppose we find the edges: $(x_1, x_2)$ and $(x_2, x_3)$. What is the edge starting at $x_3$?

Finding the aforementioned edges implies that $\gamma(x_1, x_2) < \gamma(x_1, x_3)$ and $\gamma(x_2, x_3) < \gamma(x_1, x_2)$. Because distances are symmetric, $\gamma(x_1, x_2) < \gamma(x_3, x_1)$ and $\gamma(x_3, x_2) < \gamma(x_1, x_2)$.

Thus, $\gamma(x_3, x_2) < \gamma(x_1, x_2) < \gamma(x_3, x_1)$. Henceforth the third edge of the graph must be $(x_3, x_2)$.

This counter-example shows that neighborhoods are not entirely independent. Rather, there are less possible choices than the total number of vertices in the graph as distance properties must be respected.

## D  WHITE NOISE SWEEP

We further analyze the effects of changing the noise level by using a sweep, as follows. We fixed a neighborhood value $k = 20$ and evaluated NNGS$(X, Y, k)$ for various SNR values. The SNR sweep shown in Figure 9 indicates that NNGS$(X, Y, k)$ saturates in SNRs more extreme than -10 dB and 30 dB. This behavior is also found in Figure 1, which shows that the curves outside these bounds are very close to the extremes NNGS$(X, Y, k) = 1$ and NNGS$(X, Y, k) = H(k)$. This is an important observation, as it determines the limits within NNGS$(X, Y, k)$ can be effective.

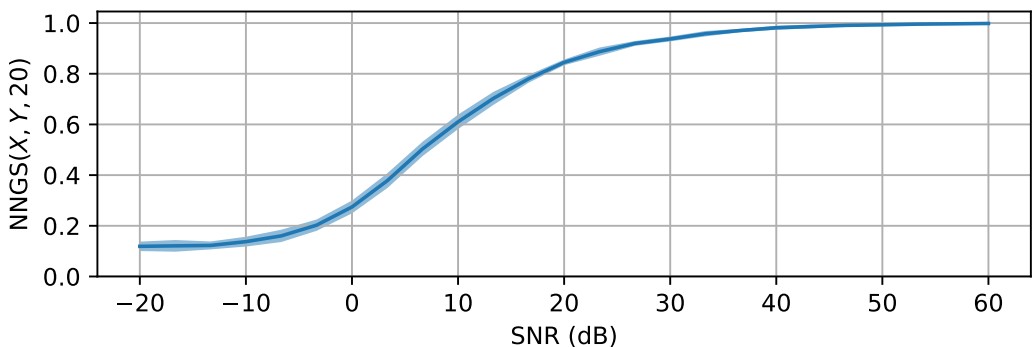

Figure 9: Similarity $\text{NNGS}(X, Y, k)$ as a function on the SNR used to distort the point cloud. Higher noise levels (lower SNR) lead to lower similarity, while lower noise levels (higher SNR) lead to higher similarity. Importantly, the measure saturates with SNR lower than -10 dB or higher than 30 dB.

## E   ALGORITHM AND PARAMETERS USED TO CREATE THE BLOBS IN SECTION 3.3

The variables used in the CreateAlignedDataset procedure are:

- n_dim: integer. Number of dimensions for the aligned datasets
- n_items: list of integers. Number of items per data blob.
- $\mu_1$ and $\mu_2$: list of values. Blob centers for each data blob.
- $\sigma_1$ and $\sigma_2$: list of values. Blob spreads for each data blob.
- noise: float. Standard deviation of noise added to each data blob.

---
**Algorithm 1** Create Aligned Dataset

---
1: **procedure** CREATEALIGNEDDATASET($n\_dim, n\_items, \mu_1, \sigma_1, \mu_2, \sigma_2, noise$)
2:     $X_1 \leftarrow$ empty array
3:     $X_2 \leftarrow$ empty array
4:     **for** $i = 0$ to $\text{len}(n\_items) - 1$ **do**
5:         $x \leftarrow$ random normal distribution$(0, 1, (n\_items[i], n\_dim))$
6:         $x_1 \leftarrow x \times \sigma_1[i] + \mu_1[i]$
7:         $x_2 \leftarrow x \times \sigma_2[i] + \mu_2[i]$
8:         $\phi \leftarrow$ random normal distribution$(0, 1, (n\_items[i], n\_dim)) \times noise[i]$
9:         $x_2 \leftarrow x_2 + \phi$
10:         Append $x_1$ to $X_1$ as a new line
11:         Append $x_2$ to $X_2$ as a new line
12:     **end for**
13:     **return** $X_1, X_2$
14: **end procedure**

---

|        | Blobs w/ Dif. Scales | Unbal. Blobs w/ Dif. Scales | Noise Within Blobs | Shuffled Blobs |
|--------|--------------------|--------------------|--------------------|--------------------|
| n_dim  | 20                 | 20                 | 20                 | 20                 |
| n_items | 200, 200          | 10, 1000           | 100, 100, 100, 100 | 100, 100, 100, 100 |
| mu1    | -1, 3              | -1, 3              | 0, 1, 2, 3         | 0, 1, 2, 3         |
| sigma1 | 3, 0.1             | 3, 0.1             | 0.1, 0.1, 0.1, 0.1 | 0.1, 0.1, 0.1, 0.1 |
| mu2    | -1, 3              | -1, 3              | 0, 1, 2, 3         | 2, 1, 3, 0         |
| sigma2 | 0.1, 3             | 0.1, 3             | 0.1, 0.1, 0.1, 0.1 | 0.1, 0.1, 0.1, 0.1 |
| noise  | 0, 0               | 0, 0               | 0.5, 0.5, 0.5, 0.5 | 0, 0, 0, 0         |

Table 2: Parameters for experiments perfomred in Section 3.3.

# F    LIST OF PROMPTS USED IN SECTION 4.2

- a photo of a X.
- a blurry photo of a X.
- a black and white photo of a X.
- a low contrast photo of a X.
- a high contrast photo of a X.
- a bad photo of a X.
- a good photo of a X.
- a photo of a small X.
- a photo of a big X.
- a photo of the X.
- a blurry photo of the X.
- a black and white photo of the X.
- a low contrast photo of the X.
- a high contrast photo of the X.
- a bad photo of the X.
- a good photo of the X.
- a photo of the small X.
- a photo of the big X.
- not a photo of a X.
- not a blurry photo of a X.
- not a black and white photo of a X.
- not a low contrast photo of a X.
- not a high contrast photo of a X.
- not a bad photo of a X.
- not a good photo of a X.
- not a photo of a small X.
- not a photo of a big X.
- not a photo of the X.
- not a blurry photo of the X.
- not a black and white photo of the X.
- not a low contrast photo of the X.
- not a high contrast photo of the X.
- not a bad photo of the X.
- not a good photo of the X.
- not a photo of the small X.
- not a photo of the big X.
- Luke. I am your X.
- The sound of a X.
- Feel the power of a X.
- Born to be X.
- I am feeling supersonic, give me X and tonic.
- Stop trying to make X happen!
- Perhaps X could help us save Robin from The Joker.
- The wheels on the X go round and round.
- We all live in a yellow X
- I can find the X here.
- X, we have a problem.
- You cannot handle the X.
- I am the X of the world.
- There is no place like X.
- You are a X, Harry!
- AND MY X!
- We must take the X to Mordor!

