# OpenReview forum: "Measuring similarity between embedding spaces using induced neighborhood graphs"
_ICLR.cc/2025/Conference — Submitted to ICLR 2025_

### Official Review · Reviewer_e5Rj · 2024-11-02

**Soundness:** 2
**Presentation:** 2
**Contribution:** 2
**Rating:** 5
**Confidence:** 3

**Summary:**

This paper proposes a new metric, named NNGS, to evaluate the similarity between paired item representations. NNGS is based on the structural similarity between the nearest-neighbors of induced graphs. Two case studies in analogy and zero-shot classification tasks demonstrate the effectiveness of the proposed method.

**Strengths:**

1. This paper introduces a new metric that assesses the similarity between paired item representations by examining the structural similarity within the neighborhoods of induced graphs.
2. The effectiveness of the proposed method is demonstrated in two scenarios: analogy tasks and zero-shot classification tasks.

**Weaknesses:**

1. The methods discussed in the related work section are insufficient. For instance, only GloVe and CLIP are mentioned for semantic word embeddings and cross-modal embeddings, respectively. A broader range of relevant methods should be included, such as BLIP and Flamingo, to provide a more comprehensive review.
2. The quality of the figures does not meet the standards expected at top conferences like ICLR. For example, Figures 5 and 6 contain overlapping text, which significantly impacts readability and clarity.
3. The comparison is limited to CKA, a method proposed in 2019. To strengthen the evaluation, more recent methods should be included for comparison.

**Questions:**

None

---

> ### Author Response · Authors · 2024-11-26
> **Thanks for your review!**
>
> Dear reviewer e5Rj
>
> Thanks for the effort put into reviewing our manuscript. Regarding your concerns:
>
> 1. We have added experiments using BLIP-2 embeddings, the ImageNet dataset, and GULP similarities. These further corroborate with our current findings.
> 2. We removed the text from the scatter plots. Thanks for this suggestion; the figures are much clearer now.
> 3. We added GULP as another baseline for evaluation.
>
> We hope to have addressed all your concerns, and kindly ask you for an increase in the review grade.

---

### Official Review · Reviewer_A643 · 2024-11-03

**Soundness:** 3
**Presentation:** 3
**Contribution:** 2
**Rating:** 5
**Confidence:** 3

**Summary:**

This paper proposes a novel method, namely “Nearest Neighbor Graph Similarity (NNGS)”, to evaluate the similarity between paired embedding spaces. Through case studies of the Glove and CLIP models, this paper validates the effectiveness of NNGS, showing a correlation between similarity and task-specific accuracy such as analogy calculation and cross-modal zero-shot classification. These findings can help to understand the reasons behind performance differences in these tasks and provide directions for improving the design of paired embedding models.

**Strengths:**

1.	Introduces a novel metric, NNGS, for evaluating similarity between paired embedding spaces.
2.	NNGS excels in identifying similar structures across different scales, which is challenging for traditional kernel methods like CKA.

**Weaknesses:**

1.	NNGS is based on Jaccard similarity, therefore it is non-differentiable and cannot be used for end-to-end learning of structurally similar representations, limiting its use to measuring the similarity between paired embeddings of items.
2.	This paper uses the K-nearest neighbor algorithm, which has high computational complexity when used on large-scale points, requiring high hardware computing resources.
3.	NNGS is constrained by KNN because KNN is sensitive to outliers in the data, which may affect its performance.
4.	the choice of k is crucial, and in high-dimensional spaces, the performance of KNN may also deteriorate. The changing the value of k implies in a modification of the locality of the similarity measure. Although NNGS demonstrates good performance, there is a lack of detailed discussion on how to select the value of k and other parameters.
5.	Figure 1 illustrates the relationship between NNGS and the number of selected k-nearest neighbors, indicating that NNGS increases as k increases; this contradicts the comparison with the last two columns of Table 1.
6.	In Line 290, the reference to Table 1 is incorrectly labels as Table 3.3.

**Questions:**

1.	In Figure 1, the relationship expressed on the same curve represents the variation of NNGS with the number of selected k nearest neighbors, indicating that as the number of k nearest neighbors increases, NNGS also increases. This contradicts the data in the last two columns of the first row in Table 1. Could you please explain this discrepancy?

2.	From Equation 7, it can be seen that c = k/(n - 1),  where k is controlled by c. So, how to choose c?

3.	In section 3.1, how is the existence of a one-to-one correspondence proven in real-world scenarios? Can the authors provide relevant theoretical or empirical support for this assumption?

---

> ### Author Response · Authors · 2024-11-26
> **Thanks for your review!**
>
> Dear reviewer A643
>
> Thanks for the effort put into reviewing our work. We have addressed your questions as follows:
>
> 1. This discrepancy was found because the datasets were created differently. As discussed in Section 3.3 (and Appendix D), data for each row in Table 1 is consisted of data grouped within blobs (clusters) with a particular added noise, whereas data for Table 1 consists of a unique dataset with all data in a single cluster. The process to create each of the datasets are discussed in Section 3.2.2 (for Figure 1) and Section 3.3 (for Table 1).
>
> 2. The parameter c is an evidence that larger datasets could use proportionally larger neighborhood sizes to generate the same parameters. Different values of c relate to different values of k and should be chosen according to the desired neighborhood size for analysis, as we clarify in Section 3.3.
>
> 3. A one-on-one correspondence is found in paired multimodal datasets used to train contrastive learning machines such as CLIP. We have added this as a footnote for quicker reference.
>
> Also, we have corrected the reference to Table 1. Regarding weakness 5, we clarify that this is because Figure 1 uses a single-cluster dataset where the only distortion is additive noise, whereas the results in Table 1 regard datasets with two clusters, hence there are distortions caused by unbalance, noise in cluster positioning, and noise within each cluster. This discussion was added to the manuscript.
>
> We hope to have addressed all your concerns, and kindly ask you for an increase in the review grade.

---

### Official Review · Reviewer_cKTW · 2024-11-03

**Soundness:** 2
**Presentation:** 3
**Contribution:** 2
**Rating:** 5
**Confidence:** 5

**Summary:**

The paper introduces a new metric, Nearest Neighborhood Graph Similarity (NNGS), designed to evaluate the similarity between embedding spaces by examining the structural similarity of induced neighborhood graphs. The authors use Jaccard similarity to assess overlap between nearest neighbors in paired embeddings. In doing so, NNGS enables comparisons across domains or modalities (e.g., text and image). NNGS is validated on two experimental case studies using analogies and GloVE embeddings, as well as CLIP embeddings on CIFAR100.

**Strengths:**

Generally, the paper is structured well, which as a result makes it fairly straighforward to follow the train of throught.
Moreover, I believe that the authors did a good job providing some crucial theoretical fundamentals in support of NNGS across section 3, which offers some interesting insights on respective measure bounds, e.g. for two independent point clouds and variations thereof.

**Weaknesses:**

Unfortunately, I am currently strongly advocating for rejection, as the paper as several large issues alongside lack of clarity in different important parts. I have ordered these based on their importance to me.

One major problem with this work is the lack of meaningful experiments, both with respect to their setup, as well as their breadth and depth.

* Testing a metric on two small-scale case studies is simply insufficient. To understand if NNGS holds any relevant benefits over NNGS, a much larger array of tests should be conducted. What happens when the metric is applied e.g. on CLIP, but with a more out-of-distribution dataset? How do insights hold when moving to larger, higher-dimensional variants? How does it transfer to other language-based similarity tasks? And for the one application to GloVe embeddings, the authors report a single correlation value, without any additional visualization, explanation of discussion of differences.

* Moreover, the application of CLIP on just CIFAR-100 is insufficient. For one, CIFAR-100 operates in much lower resolution than the training data used for CLIP. While still applicable, insights do not necessarily transfer. At the same time, the experimental design is problematic; relying on CLIPs sensitivity to template changes can fall victim to the known bag-of-words nature in CLIP (c.f. e.g. Yuksekgonul et al. 2022, https://arxiv.org/abs/2210.01936). At the same time, the differences in correlation to CKA, particularly given that experiments were conducted on just one dataset, are insignificant.

* The authors insufficiently compare and contrast against other similarity measures (of which there are many) - both in their discussion of related works, and more importantly, their experimental case studies. Why simply focus on CKA, when SVCCA, PWCCA (Morcos et al. 2018, https://arxiv.org/pdf/1806.05759), ContraSim (Rahamim et al. 2023, https://arxiv.org/pdf/2303.16992), Brain-Score, RSA or metrics like GULP (https://arxiv.org/pdf/2210.06545) or a simple mean cosine similarity between clusters are all possible similarity measures to relate against performance changes.

In turn, this makes it unclear why NNGS should be preferred over other distance-based similarity measures. This also holds when looking at the theoretical motivation: there is no free lunch. By avoiding the explicit reliance of point distances, NNGS in return disregards relative relations between neighbouring points in a k-Neighbourhood, which in turn raises the question on whether this is a desired property or not.

* The authors for example note that "NNGS has two additional parameters when compared to CKA: the neighborhood size k, and the distance metric used to induce the neighborhood graph. In datasets with more than one cluster, these parameters can be manipulated to find similarities at different scales and in different situations." But in CKA, one can simply adjust the utilized kernel to adapt to different situations. Moreover, isn't the dependence on different scales and situations used as the motivating factor for NNGS over CKA, as e.g. noted in L108-111? It is not entirely clear to me why NNGS should, again, be preferred?

Similarly, there are several other elements of the provided motivation that remain unclear to me:

* L328: "Although theoretically the value of σ in CKA with an RBF kernel plays the same role, it is easier to find suitable values for k than to σ. >>> But why? This is very handwavy, arbitrary reasoning. Sigma is simply a hyperparameter to tune like k.

* L329: "In special, we observe that even very small values of σ were innefective to find local changes in the point clouds." >>> But in these cases, CKA does allow for simple changes in the underlying kernel to much better account for particular point cloud structures, no?

Finally, the actual novelty is also limited: The Jaccard similarity is exactly defined as a distance over vertices' graph neighbourhoods. The singular novelty in this work is thus the application to a graph induces by some distance function; with works such as Sobal et al. 2024 (https://arxiv.org/pdf/2407.18134) having already investigated the use of distance-induced graphs for e.g. contrasive learning.

Also a smaller, nitpicky issue: The paper uses in parts weird formulation throughout, such as "datablob"; which I assume refers to clusters?

**Questions:**

See weaknesses. I am currently strongly advocating for rejection; but would be willing to raise my score if the authors significantly extend their experimental study, and provided convincing arguments regarding the novelty of the proposed NNGS.

---

> ### Author Response · Authors · 2024-11-26
> **Thanks for your comments!**
>
> Dear reviewer cKTW,
>
> We immensely thank you for the effort put into the reviewing process. We have considered all your concerns, and acted on them as follows:
>
> > Testing a metric on two small-scale case studies is simply insufficient.
> > Moreover, the application of CLIP on just CIFAR-100 is insufficient.
>
> Thanks for this kind suggestion. We have added experiments using:
> * GULP as an additional baseline
> * ImageNet as an additional dataset
> * BLIP-2 as an additional method to generate multimodal embeddings.
>
> We increased the experimental report so that it now contains the visualization related to all tested approaches (NNGS, CKA, and GULP). Each experiment is now reported with visualizations accompanying the numeric results.
>
> > The authors insufficiently compare and contrast against other similarity measures (of which there are many)
>
> We thank you for this review. We addressed this concern by adding GULP to our benchmark. We also considered ContraSim, but, as it this is a learned measure, the number of necessary configurations to compare performance would be too high - in fact, any function could theoretically be approximated by ContraSim with adequate data and with large enough encoders.
>
> > In turn, this makes it unclear why NNGS should be preferred over other distance-based similarity measures.
>
> As we have discussed above, we do not with to claim to have an overall "better" metric. Rather, we show that NNGS is more effective and easier to interpret in the specific use cases we work with. The definition of "similarity" is not a general consensus over the literature, and each measure is built upon different assumptions regarding that. Hence, we now clarify in our conclusion that "different similarity metrics can work harmoniously to provide different viewpoints about embedding spaces"
>
> > It is not entirely clear to me why NNGS should, again, be preferred?
>
> Thanks for this note.
>
> In Table 1, we show that sigma can be adjusted in RBF-CKA, but it only relates to proximity if clusters have a more or less uniform composition. In our experiments, we were unable to tune RBF-CKA to differentiate noise within data clusters from shuffling data clusters themselves. However, finding the value of k only requires looking at the dataset and choosing which is more interesting for the specific use case, that is, k is directly related to the data composition and easier to interpret.
>
> As for the question: "why NNGS should be preferred", the discussion is more profound. NNGS, CKA, GULP, and each other similarity measure accounts for different aspects of what could be defined as similarity in embeddings. In special, we note that NNGS comes from defining similar embeddings as those in which neighborhoods are preserved, while GULP defines similar embeddings as those that lead to a similar error in a linear prediction. These underlying definitions are so different that it makes little sense to state that one should be preferred over the other a priori. However, if we know that what we wish to measure is the neighborhood similarity between paired representations of items, then NNGS should be preferred; likewise, if the problem at hand requires analyzing the prediction error of embeddings, then GULP should be preferred.
>
> > L328
>
> As we have clarified, k is a parameter that depends only on the number of elements in each data cluster, whereas sigma depends on the in-cluster variance and the between-cluster distances. Hence, k can be immediately set and changed to tune NNGS to bring information on more local or more global scales.
>
> > L329:
>
> The proposed kernels in the original paper are linear and RBF. It is likely that, theoretically, we could find a specific kernel for each dataset. However, this falls out of the scope of this paper, as our work is about using the
>
> > Novelty
>
> Thanks for the reference. We note that our work was first submitted (and rejected) to Neurips 2024 in May 14th, as will soon be available in OpenReview, which precedes the first version of Sobal et al.'s preprint; however, we have added Sobal et al. as a reference in the present work.
>
> Their work use a soft neighborhood approach to train a multimodal machine similar to CLIP and find a minor improvement in a downstream supervised classification process. We have provided a test in the same context (contrastive multimodal learning) that shows NNGS correlates to Zero-Shot classificatio performance, which was not approached by Sobal et al.'s work. Also, we note that we present a similarity measure to compare embeddings, not a loss function for contrastive learning, which can be used in other contexts. Additionally, we note that the fact that a similar idea lead to a higher accuracy in a downsteam task is further evidence that NNGS can give useful insight regarding embedding spaces.
>
> We thank you for the notes that have helped increasing the quality of our work. We hope to have addressed all your concerns, and kindly ask you for an increase in the review grade.

---

> > ### Comment · Reviewer_cKTW · 2024-12-02
> > **Response to Rebuttal**
> >
> > I thank the authors for their thorough rebuttal, which has addressed several points I raised in my review:
> >
> > * Positioning NNGS much more as an additional metric tool alongside CKS or GULP as opposed to a replacement,
> > * inclusion of GULP, ImageNet and BLIP-2 as additional metric, dataset and embedder for comparison, respectively,
> > * and discussion of better interpretability of NNGS.
> >
> > The authors did a great job incorporating a much more comprehensive experimental study, and as a result, I have raised my score from 3 to 5. Unfortunately, I am still slightly advocating for rejection, because:
> >
> > * While it is really great to see ImageNet added as an additional benchmark dataset, this only leaves two small-scale studies (Glove & Cifar100) and ImageNet. While older works such as CKA only study Cifar10/100, CKA has been validated and applied to many more datasets over the past years. Incorporating more datasets is in my eyes therefore essential, particularly as the
> > * my review listed other metrics (which was not at all meant as a comprehensive list) such as SVCCA, PWCCA, Brain-Score or RSA. While I understand why ContraSim is not a suitable comparison to make, what about comparison other metrics such as SVCCA or PWCCA? And if no comparison are made, following the discussion of ContraSim in the rebuttal, it's crucial to highlight why that is the case.

---

### Meta-Review · Area_Chair_ge8k · 2024-12-20

**Metareview:**

This submission proposes Nearest Neighbor Graph Similarity (NNGS) as a metric for evaluating similarity between embedding spaces by examining structural similarities within neighborhoods of induced graphs. While revisions during rebuttal addressed some concerns, significant issues remain, particularly in terms of novelty, experimental validation, and breadth of comparisons. Despite the added experiments, the scope remains limited, particularly for a metric that aims to generalize across embeddings and modalities.

**Additional Comments On Reviewer Discussion:**

Most reviewers are still negative after the rebuttal and discussion.

Reviewer cKTW (Score: 5): While acknowledging improvements, raised ongoing concerns about experimental scope and lack of diverse comparisons, advocating for rejection.

Reviewer A643 (Score: 5): Criticized parameter sensitivity, computational inefficiency, and limited experimental breadth, maintaining a score below acceptance.

Reviewer e5Rj (Score: 5): Highlighted insufficient related work and comparisons, agreeing that the experimental section does not adequately validate the proposed metric.

---

### Decision · Program_Chairs · 2025-01-22

Reject